# The Impact of Sustainable Exercise and the Number of Pregnancies on Self-Efficacy, Self-Esteem, and Assertiveness Levels in Pregnant Women

**Eren Uluoz** [1] , **Turhan Toros** [2,*] , **Emre Bulent Ogras** [3] , **Cenk Temel** [4] , **Cihat Korkmaz** [5] , **Muzaffer Toprak Keskin** [6] **and Ibrahim Efe Etiler** [3]

1    Faculty of Sport Sciences, Çukurova University, Adana 01000, Turkey; proferde@hotmail.com
2    Department of Coaching Education, Mersin University, Mersin 33000, Turkey
3    Faculty of Sport Sciences, Mersin University, Mersin 33000, Turkey; emrebulentogras@gmail.com (E.B.O.); efeetiler@gmail.com (I.E.E.)
4    Department of Sport Management, Akdeniz University, Antalya 07070, Turkey; cenktemel@akdeniz.edu.tr
5    Faculty of Sport Sciences, Kahramanmaras Sutcu Imam University, Kahramanmaras 46000, Turkey; korkmaz.cihat68@gmail.com
6    Faculty of Sport Sciences, Nevsehir Hacı Bektas Veli University, Nevsehir 50000, Turkey; toprakkeskin@hotmail.com
*    Correspondence: turhantoros@yahoo.com

**Abstract:** This study examined the variations in self-efficacy, self-esteem, and assertiveness levels among pregnant women engaging in sustainable exercise compared to those performing no physical activity. The study also explored the connection between these changes and the number of pregnancies. The sample included 220 pregnant women engaging in sustainable exercise and 210 pregnant women performing no physical activity. Sustainable exercisers were chosen from those engaged in physical activity for at least 30 min, twice a week. The participants were in the fourth to seventh month of their pregnancy. A simple random sampling technique was used to choose participants and a total of 430 pregnant women volunteered to participate in the study. The mean age of the participants was $31.45 \pm 12.11$ years. Data collection tools were the Self-Efficacy Scale (SES), the Coopersmith Self-Esteem Inventory (CSEI), and the Rathus Assertiveness Schedule (RAS). In data analysis, the impact of independent variables on self-efficacy, self-esteem, and assertiveness was evaluated by one-way ANOVA in groups of more than two, $t$-test in paired groups, the relationship between some independent variables and scales was evaluated by correlation, and descriptive features were shown as percentages. In cases where variance analyses were significant at 0.05 ($p < 0.05$), Tukey's test was used as a post hoc test. The study's results indicated a significant disparity between the mean self-efficacy and self-esteem scores of women engaged in sustainable exercise compared to those who were not. However, there was no significant difference between the two groups in terms of assertiveness levels. The mean scores of self-efficacy, self-esteem, and assertiveness differed significantly in respect to the number of pregnancies in exercising women. However, there were no significant differences in mean scores of self-efficacy, self-esteem, and assertiveness scores in terms of the number of pregnancies in women who did not exercise.

**Keywords:** sustainable exercise; pregnant women; self-efficacy; assertiveness; self-esteem





## 1. Introduction

Sustainability, typically defined as an ecosystem's capacity to maintain its health and integrity without exhausting crucial resources [1], represents a multifaceted concept. It encompasses judicious utilization of natural resources, minimizing waste, recycling, and addressing future generations' needs with a focus on environmental protection. An all-encompassing evaluation of the long-term implications of resource consumption is

paramount. The principal objective of sustainability is to safeguard and enhance human capital, necessitating significant investment in areas like health and education. This investment ensures access to essential services, proper nutrition, knowledge, and skill-building opportunities. In the business context, an organization ought to perceive itself as a vital societal component, endorsing values that put a premium on human assets [2].

Physical activity and regular exercise offer numerous health advantages at both personal and societal levels. Emphasizing individual physiological and behavioral characteristics, the promotion of physical activity seeks to foster an active lifestyle. The concept of 'active living,' which embeds physical activity into everyday routines to stimulate a more active lifestyle, is pivotal in this context [3]. This behavior is instrumental in augmenting physical activity and exercise. Sustainable physical activity, a fundamental element of sustainable living, bolsters the sustainable development of individuals and society [4]. Both active and sustainable living principles root themselves in daily life, necessitating the incorporation of physical activity and sustainable practices, respectively. The individual serves as the primary nexus in the chain of a sustainable economy, society, and environment [5]. Research indicates key principles for sustainable physical activity, including setting realistic expectations, being cognizant of desired physical, mental, and emotional states, establishing attainable goals, objectively monitoring progress, revising the plan if goals are not met, and integrating exercise as an integral part of lifestyle rather than as a mere ancillary need [6–9].

Bandura (1977) introduced the concept of self-efficacy as part of 'cognitive behavioral modification' [10]. This term refers to an individual's confidence in their ability to accomplish goal-directed behaviors. Self-efficacy plays a pivotal role in human behavior, influencing the actions we take, the effort we invest, and our persistence in problem-solving tasks [11]. Self-efficacy derives from four intertwined sources: performance accomplishments, vicarious experiences, verbal persuasion, and emotional states. Performance accomplishments, predicated on personal learning experiences, represent the most potent source. Successful experiences fortify learning expectations, while recurrent failures may impede effective learning. These experiences can offer invaluable feedback to enhance performance. The observation of successful models can also assist in skill acquisition, though it is not as influential as personal experiences [12].

Self-esteem embodies an individual's sense of self-satisfaction, untethered to feelings of superiority or inferiority compared to others. It encompasses self-worth, a favorable self-image, and self-appreciation [13]. Shaped by life events from birth to adolescence, self-esteem consists of emotional, mental, physical, and social elements. Crucial factors influencing its development include a sense of self-value, the effective use of personal talents and knowledge, societal acceptance, and contentment with physical attributes [14]. Serving as a bedrock for resilience, self-esteem is intrinsically tied to psychological health and general well-being [15,16]. It represents a state of self-acceptance founded on personal self-assessment, irrespective of any prized traits or qualities.

Social skills, particularly assertiveness, significantly impact the quality of communication and interaction [17]. Assertiveness, a constructive behavior, characterizes individuals who are self-assured, respectful of others' rights, and can articulate their thoughts and feelings effectively [18]. Salter, a trailblazer in assertive behavior therapy, offered training to individuals grappling with forthright and unambiguous emotional expression. He co-developed a model with Wolpe aimed at enhancing self-expression [19]. There is a positive correlation between assertiveness and self-confidence, indicating that the development of one could foster the growth of the other. Assertive individuals often display traits such as the accurate identification and expression of emotions, ethical goal setting, autonomous decision-making, clear and confident communication, and considerate behavior to prevent offending oneself or others [20].

Physical activity, a key component of a healthy lifestyle, refers to structured exercise designed to enhance physical fitness [21]. Pregnancy often serves as a catalyst for women to adopt healthier lifestyles and offers the chance to monitor physical activity due to frequent



medical consultations. Regular exercise during pregnancy can mitigate conditions such as gestational diabetes, hypertension, operative delivery, excessive postpartum weight gain, and postpartum depression [22,23]. Nevertheless, a lack of information or apprehensions about potential risks may deter the commencement of exercise. Pregnancy-induced physiological adaptations, such as shifts in heart rate and body positioning, can influence exercise capacity and type [24]. Ideally, exercise should begin after the first trimester and continue regularly until birth. Considering individual variations, each pregnant woman should consult with a physical therapist to customize an exercise program that suits their specific fitness levels and needs. Pregnancy, a particularly emotional period in a woman's life, requires heightened physical and emotional self-care [25,26]. Regular exercise, beyond maintaining physical health, can boost psychological well-being, self-efficacy, self-esteem, and self-confidence in expectant mothers [27–29]. Furthermore, it acts as a vital instrument to manage weight gain during pregnancy.

Physical activity during pregnancy brings multiple benefits, such as diminished labor pain, fewer medical interventions, and quicker postpartum recovery [30,31]. Regular exercise also positively impacts pregnant women's self-efficacy, self-esteem, and assertiveness [32,33]. The objective of this study is to explore the variations in self-efficacy, self-esteem, and assertiveness levels among pregnant women who engage in exercise versus those who do not. The study also seeks to understand how these changes correlate with the number of pregnancies.

In line with the aim of the research, answers to the following research problems were sought:

1.  Do the self-efficacy, self-esteem and assertiveness levels of pregnant women who exercise differ significantly from those who do not exercise?
2.  Do the self-efficacy levels of pregnant women who exercise sustainably differ significantly in terms of the number of pregnancies?
3.  Do the self-efficacy levels of pregnant women who do sustainable exercise differ significantly in terms of number of pregnancies?
4.  Do the assertiveness levels of pregnant women who do sustainable exercise differ significantly according to the number of pregnancies?
5.  Do the self-efficacy levels of pregnant women who do not engage in sustainable exercise differ significantly according to the number of pregnancies?
6.  Do the self-esteem levels of pregnant women who do not engage in sustainable exercise differ significantly according to the number of pregnancies?
7.  Do the assertiveness levels of pregnant women who do not engage in sustainable exercise differ significantly according to the number of pregnancies?

## 2. Materials and Methods

The study sample comprised 430 pregnant women, of which 220 engaged in sustainable exercise and 210 did not partake in any physical activity. Participants adhering to regular exercise were selected based on their commitment to a regimen of at least two sessions of 30 min each per week. The selection occurred between the fourth and seventh months of their pregnancy. A simple random sampling method was utilized to recruit participants. The participants' average age was $31.45 \pm 12.11$ years.

### 2.1. Research Design

This study adopted a correlational survey design to explore potential associations between self-esteem, self-efficacy, assertiveness levels, and the number of pregnancies among pregnant women engaging in sustainable exercise.

Ethical Approval: This study received ethical approval from the T.C. Mersin University Sports Sciences Ethics Committee, as per the decision dated 4 April 2022, with reference number 051.

## 2.2. Data Collection Tools

**Personal Information Form:** The research team developed a personal information form to gather independent variables, including whether the participants exercised or not and their number of pregnancies.

**Self-Efficacy Scale (SES):** The study utilized the Self-Efficacy Scale (SES) by Riggs, Warka, Babasa, Betancourt, and Hooker, designed to gauge individuals' beliefs about their capabilities. Öcel adapted the scale to Turkish in 2002, comprising ten items [34]. Participants express their agreement with the statements on a five-point Likert scale, and a total efficacy score is generated by summing the numeric values marked for each item. The scale ranges from 10 to 50, with a higher score indicating strong self-efficacy belief. The internal consistency coefficient reported by various researchers was 0.80 [35]. Factor analysis performed on the Turkish version data confirmed its construct validity, with all item factor ranges between 0.32 and 0.85, and an internal consistency coefficient of 0.61.

**Coopersmith Self-Esteem Inventory (CSEI):** Participants' global self-esteem was measured using the CSEI, a self-report questionnaire by Stanley Coopersmith and validated for Turkish by Turan and Tufan in 1987 [36]. The scale comprises 25 items, with test-retest reliability reported as 0.65 and 0.76 in studies conducted a year apart. Items cover aspects like life perspective, family relations, social connections, and resilience, with scores ranging from 0 to 100. A below-average score indicates low self-esteem, whereas an above-average score indicates high self-esteem.

**Rathus Assertiveness Schedule (RAS):** To assess assertiveness levels, we used the RAS, developed by Rathus and validated for Turkish by Voltan [37], who reported an alpha consistency coefficient of 0.70 and a test-retest reliability of 0.92. The scale comprises 30 items, applicable to both adolescents and adults, with 17 items negatively phrased and 13 positively phrased to prevent response bias. Scores below '+10' suggest timidity, while scores above '+10' suggest assertiveness.

## 2.3. Data Analysis

In the data analysis phase, the effects of independent variables on self-efficacy, self-esteem, and assertiveness levels were examined using One-Way ANOVA for groups of more than two, and an independent samples *t*-test was used for pairs of groups. Descriptive characteristics were presented in terms of percentages and frequencies. In cases where the variance analysis was significant at the 0.05 level ($p < 0.05$), Tukey's post hoc analysis was applied.

## 2.4. Results

The results showed a significant difference in the mean self-efficacy levels of exercisers and non-exercisers ($p < 0.05$), as women engaged in sustainable exercise enjoyed much greater self-efficacy than non-exercisers. Significant differences were found between the mean self-esteem scores of exercisers and non-exercisers ($p < 0.05$). This difference stemmed from greater self-esteem among exercising pregnant women as compared to non-exercisers. However, there were no significant differences in the mean assertiveness scores between the two groups ($p > 0.05$) (Table 1).

Analysis of research data revealed a significant difference in the mean self-esteem scores of women who exercised regularly ($p < 0.05$). Tukey's test was used to determine whether there was a significant difference between the groups, and the mean self-esteem scores of women exercising during their first pregnancy were found to be significantly higher than those of women in their fourth pregnancy and beyond (Table 2).

**Table 1.** Distribution of Self-Efficacy, Self-Esteem, and Assertiveness Scores in Pregnant Women Exercising or Not Exercising.

| | *N* | Self-Efficacy (Mean ± SD) | Self-Esteem (Mean ± SD) | Assertiveness (Mean ± SD) |
|---|---|---|---|---|
| Pregnant women exercising | 220 | 3.85 ± 1.77 | 77.36 ± 16.09 | 16.69 ± 2.82 |
| Pregnant women with no exercise | 210 | 1.76 ± 1.36 | 71.89 ± 15.47 | 22.72 ± 0.13 |
| *N* | 430 | $t = 2.112$ $p = 0.035$ | $t = 2.334$ $p = 0.026$ | $t = -1.349$ $p = 0.231$ |

SD: standard deviation.

**Table 2.** Difference Analysis between Number of Pregnancies and Self-Esteem in Pregnant Women Engaging in Sustainable Exercise.

| Pregnant Women | *N* | M | SD | *F* | *p* | Significant Difference |
|---|---|---|---|---|---|---|
| First pregnancy | 95 | 77.77 | ±16.09 | | | |
| Second pregnancy | 87 | 74.79 | ±16.12 | 2.476 | 0.022 | 1 > 4 |
| Third pregnancy | 27 | 71.63 | ±14.34 | | | |
| Fourth pregnancy and above | 11 | 68.14 | ±15.68 | | | |

A significant difference was found in the mean self-efficacy scores in terms of the number of pregnancies of pregnant women who practiced sustainable exercise ($p < 0.05$). Tukey's test was performed to determine between which groups there was a significant difference. Analysis showed that the mean self-efficacy scores of women who practiced sustainable exercise and were in their first pregnancy were significantly higher than the mean self-efficacy scores of women who were in the fourth pregnancy and above (Table 3).

**Table 3.** Difference Analysis between Number of Pregnancies and Self-Efficacy in Pregnant Women Engaging in Sustainable Exercise.

| Pregnant Women | *N* | M | SD | *F* | *p* | Significant Difference |
|---|---|---|---|---|---|---|
| First pregnancy | 95 | 3.98 | ±1.65 | | | |
| Second pregnancy | 87 | 2.95 | ±1.43 | 2.708 | 0.018 | 1 > 4 |
| Third pregnancy | 27 | 2.05 | ±1.38 | | | |
| Fourth pregnancy and above | 11 | 1.12 | ±1.60 | | | |

There was a significant difference in mean assertiveness scores in terms of the number of pregnancies among pregnant women who practiced sustainable exercise ($p < 0.05$). Tukey's test was performed to determine whether there was a significant difference between groups. The mean assertiveness scores of women who practiced sustainable exercise and were in their first pregnancy were significantly higher than those who were in their fourth pregnancy or above (Table 4).

**Table 4.** Difference Analysis between the Number of Pregnancies and Assertiveness in Pregnant Women Engaging in Sustainable Exercise.

| Pregnant Women Engaging in Sustainable Exercise | *N* | **M** | **SD** | *F* | *p* | Significant Difference |
|---|---|---|---|---|---|---|
| **First pregnancy** | 95 | 18.69 | ±2.01 | | | |
| **Second pregnancy** | 87 | 16.12 | ±0.95 | 2.428 | 0.023 | 1 > 4 |
| **Third pregnancy** | 27 | 14.34 | ±0.78 | | | |
| **Fourth pregnancy and above** | 11 | 12.71 | ±6.18 | | | |

No significant differences were found in mean self-esteem scores in terms of the number of pregnancies in pregnant women who did not exercise ($p > 0.05$) (Table 5).

**Table 5.** Difference Analysis between the Number of Pregnancies and Self-Esteem of Pregnant Women Not Exercising.

| Pregnant Women Who Don't Exercise | *N* | **M** | **SD** | *F* | *p* |
|---|---|---|---|---|---|
| **First pregnancy** | 102 | 71.35 | ±16.35 | | |
| **Second pregnancy** | 70 | 71.07 | ±16.24 | 1.445 | 0.116 |
| **Third pregnancy** | 24 | 71.02 | ±14.87 | | |
| **Fourth pregnancy and above** | 14 | 71.01 | ±15.40 | | |

The mean self-efficacy scores did not differ significantly in terms of the number of pregnancies among pregnant women who did not exercise ($p > 0.05$). (Table 6).

**Table 6.** Difference Analysis Between Number of Pregnancies and Self-Efficacy in Pregnant Women Not Exercising.

| Pregnant Women Who Don't Exercise | *N* | **M** | **SD** | *F* | *p* |
|---|---|---|---|---|---|
| **First pregnancy** | 102 | 2.15 | ±1.98 | | |
| **Second pregnancy** | 70 | 2.10 | ±1.65 | | |
| **Third pregnancy** | 24 | 2.05 | ±1.19 | | |
| **Fourth pregnancy and above** | 14 | 2.05 | ±1.02 | 1.711 | 0.123 |

Analyses indicated no significant difference in the mean assertiveness scores of the non-exercising women in relation to the number of pregnancies they had ($p > 0.05$). (Table 7).

**Table 7.** Difference Analysis Between Number of Pregnancies and Assertiveness in Pregnant Women Not Exercising.

| Pregnant Women Not Exercising | *N* | **M** | **SD** | *F* | *p* |
|---|---|---|---|---|---|
| **First pregnancy** | 102 | 18.43 | ±2.34 | | |
| **Second pregnancy** | 70 | 18.37 | ±0.98 | 1.134 | 0.309 |
| **Third pregnancy** | 24 | 18.07 | ±0.99 | | |
| **Fourth pregnancy and above** | 14 | 18.05 | ±4.55 | | |

## 3. Discussion

The primary aim of this study was to explore the impact of sustainable exercise and the number of pregnancies on the self-efficacy, self-esteem, and assertiveness levels of pregnant women. Our findings revealed a significant discrepancy in the mean self-efficacy scores

between pregnant women who regularly engaged in sustainable exercise and those who did not participate in physical activities. Self-efficacy encompasses several aspects such as self-confidence, belief in success, and problem-solving skills [10,11]. The mechanism by which sustainable exercise affects self-efficacy could be multifaceted. Physical activity may promote a sense of accomplishment, enhancing perceived competence and thereby boosting self-efficacy. Additionally, the physiological changes associated with regular exercise, such as improved mood and reduced stress, could indirectly influence self-efficacy by fostering a more positive mindset [38]. Furthermore, pregnant women with high self-efficacy levels can better navigate the physical and psychological challenges associated with pregnancy. The impact of feasible exercise on self-efficacy levels corroborates our findings. A previous study reported higher self-efficacy levels among women who exercised during pregnancy [38]. Moreover, another study suggested an improvement in the physical activity levels of pregnant women who received self-efficacy training [39]. This evidence further underscores the potential of exercise as a tool to enhance self-efficacy in pregnant women.

Our study data revealed a substantial difference in the mean self-esteem levels between pregnant women engaging in sustainable exercise and those who did not, with the former group displaying higher self-esteem. This difference might be explained by several theoretical mechanisms suggesting that physical exercise can positively influence self-esteem. Firstly, regular physical activity can improve self-perception and body image, which are critical components of self-esteem. Exercise can induce feelings of accomplishment, and the physical changes associated with exercise, such as increased strength and endurance, can lead to improved body image and self-confidence. These elements are especially important during pregnancy, a time of significant bodily changes that can challenge a woman's self-perception. Secondly, exercise can lead to the release of endorphins, which are chemicals in the brain that act as natural mood lifters. This endorphin release can create a positive mood state, contributing to a higher sense of self-worth and thus increased self-esteem. Thirdly, the social aspect of exercise, such as group classes or team sports, can provide a sense of belonging and accomplishment, further enhancing self-esteem. Our findings align with previous research. For instance, one study found an association between regular exercise and increased self-esteem in pregnant women [40]. These mechanisms and associations underscore the potential benefits of sustainable exercise in bolstering self-esteem during pregnancy.

In the present study, we sought to examine the relationship between assertiveness levels and engagement in sustainable exercise. Interestingly, we did not find a significant difference between the assertiveness scores of pregnant women who exercised and those who did not, suggesting that within our sample, exercise did not directly influence assertiveness levels. This finding seems contrary to existing literature. Previous studies have reported that regular aerobic exercise can increase assertiveness, as evidenced by a study on male hockey players who underwent a 12-week regular aerobic exercise program [41]. Further studies also indicated a positive relationship between exercise and assertiveness [42,43].

It is important to consider the mechanisms by which exercise could potentially influence assertiveness. Firstly, engaging in regular exercise can lead to increased self-confidence and self-efficacy which are both closely related to assertiveness. Secondly, the social interaction involved in group exercise can provide opportunities to practice assertive communication. Finally, exercise can also help reduce anxiety and improve mood, which could indirectly boost assertiveness by reducing fear or hesitation in expressing oneself. However, the interpretation of our findings should consider the research design, sample size, and assessment tools used. Our results are valid only within the context of our study and may not be generalizable to a broader population. Further research is needed to fully understand the potential relationship between exercise and assertiveness in pregnant women, considering the multitude of influencing factors.

Significant differences were noted in the self-esteem scores of pregnant women who practiced sustainable exercise, particularly when compared across the number of preg-

nancies. There was a discernible disparity in self-esteem between women in their first pregnancy and those in their fourth pregnancy or beyond, with those in their first pregnancy reporting higher self-esteem.

Multiple factors could contribute to this finding. Women in their fourth pregnancy or beyond may face additional stressors such as the demands of caring for multiple children, economic constraints, and planning for their children's future, which could potentially impact their self-esteem negatively. The mechanisms through which exercise might enhance self-esteem are multifaceted. Firstly, physical activity has been associated with improvements in body image and self-confidence which are key components of self-esteem. Secondly, exercise can lead to feelings of mastery and accomplishment which can further boost self-esteem. Furthermore, the endorphin release associated with exercise can improve mood, which may indirectly enhance self-esteem [44]. While existing research has not directly explored the relationship between the number of pregnancies, sustainable exercise, and self-esteem, it is well established that high self-esteem can promote effective problem-solving skills. Thus, self-esteem may be crucial in predicting adherence to sustainable exercise during pregnancy. Our results suggest that interventions aimed at boosting self-esteem through exercise could be beneficial, especially since low self-esteem has been linked to various mental health problems. However, more research is needed to fully elucidate these relationships [45].

In our study, we noted significant differences in self-efficacy scores among pregnant women participating in sustainable exercise, especially when comparing the number of pregnancies. The data revealed that women in their first pregnancy exhibited higher self-efficacy scores than those in their fourth pregnancy or beyond.

Various mechanisms could explain this finding. As Bandura suggested, self-efficacy is rooted in personal mastery experiences, vicarious experiences, verbal persuasion, and physiological and affective states [46]. Therefore, women in their first pregnancy may exhibit greater self-efficacy due to several factors. Firstly, these women might engage more frequently in learning experiences to better understand pregnancy and childbirth, thereby enhancing their mastery experiences [47,48]. Additionally, they might rely more on vicarious experiences by seeking advice and learning from the experiences of others, which could boost their self-efficacy. Lastly, positive verbal persuasion, such as encouragement and affirmation from healthcare providers, friends, and family, might be more prevalent during a woman's first pregnancy, further improving her belief in her capabilities [49]. However, as pregnancies progress, the demands of caring for existing children, combined with the anticipation of additional responsibilities, could potentially dampen self-efficacy. Further research is needed to fully explore these mechanisms and their impact on self-efficacy in the context of pregnancy and sustainable exercise.

While our study shows higher assertiveness levels among women exercising during their first pregnancy compared to those in their fourth pregnancy or beyond, the underlying mechanisms for this observation warrant further discussion. Firstly, exercise is known to lead to improvements in mood and mental health, due in part to the release of endorphins, which could potentially make individuals feel more confident and assertive. Additionally, setting and achieving exercise goals can enhance a sense of self-efficacy, which is closely related to assertiveness. Secondly, social interaction inherent in many forms of exercise may provide opportunities for practicing and improving assertiveness skills. Engaging in group exercise classes or sports, for instance, can offer situations where expressing one's needs and standing up for oneself is required. Lastly, first-time pregnant women, eager to ensure the best possible health outcomes for their babies, might be particularly motivated to assert their needs and boundaries, both in exercise contexts and more broadly. In contrast, women in their fourth pregnancy or beyond may be more experienced and feel less need to assert themselves in the same way, or they may have less time and energy to devote to exercise due to existing childcare responsibilities [49]. However, more research is needed to fully understand the intricate dynamics between exercise, number of pregnancies, and

assertiveness levels. Hormonal changes during pregnancy, as well as individual personality traits, could also play a role and should be explored in future studies.

## 4. Conclusions

The results of this study indicate a significant impact of sustainable exercise on the self-efficacy and self-esteem of pregnant women, with those participating in such activities demonstrating higher mean scores. Assertiveness scores, however, did not significantly differ between the two groups.

When considering the number of pregnancies, first-time pregnant women who engaged in sustainable exercise reported higher self-esteem and self-efficacy scores compared to those in their fourth or subsequent pregnancies. The same trend was observed for assertiveness scores.

In summary, sustainable exercise appears to positively influence self-efficacy and self-esteem among pregnant women, particularly in first-time pregnancies. The effect on assertiveness scores warrants further investigation.

The study findings on the relationship between sustainable exercise, the number of pregnancies and pregnant women's self-efficacy, self-esteem, and assertiveness of pregnant women have significant implications. Based on these findings, the following recommendations can be made for future research:

a.　Future researchers may consider designs that include a larger study sample.
b.　Exercise and pregnancy are not the only factors that influence personality traits such as self-efficacy, self-esteem, and assertiveness, so other potential predictors along with these variables could be investigated.
c.　Our sample consisted of a certain age group and ethnicity. However, future research could increase the generalizability of the results by comparing the data across different age groups and diverse ethnic groups.
d.　Future studies could also include the postpartum period in their analyses. Exercise may have benefits not only during pregnancy, but also after childbirth, hence analysis of both pregnancy and postpartum period may yield interesting results.

**Author Contributions:** Conceptualization, T.T. and E.U.; methodology, E.B.O.; software, E.U.; validation, M.T.K., C.T. and C.K.; formal analysis, T.T.; investigation, E.B.O.; resources, I.E.E.; data curation, T.T.; writing—original draft preparation, M.T.K.; writing—review and editing, T.T.; visualization, E.B.O.; supervision, T.T.; project administration, T.T.; funding acquisition, C.T. All authors have read and agreed to the published version of the manuscript.

**Funding:** This research received no external funding.

**Institutional Review Board Statement:** This study received ethical approval from the T.C. Mersin University Sports Sciences Ethics Committee, as per the decision dated 4 April 2022, with reference number 051.

**Informed Consent Statement:** Informed consent was obtained from all subjects involved in the study.

**Conflicts of Interest:** The authors declare no conflict of interest.

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
