# Peer review of "The Impact of Sustainable Exercise and the Number of Pregnancies on Self-Efficacy, Self-Esteem, and Assertiveness Levels in Pregnant Women"

_sustainability, doi:10.3390/su15118978_

Round 1
Reviewer 1 Report (Previous Reviewer 4)
The authors have substantially improved all the changes proposed by the reviewers. This would allow the article to be published
Author Response
Dear Reviewer,
We woldu like to thank you for the insightful comments and suggestions. We made all possible changes that were suggested and detailed the changes in the table below. Prior to response your comments we want to inform you that all the revisions and improvements are highlighted yellow in the revised version of our manuscript. We sincerely appreciate your insightful comments on our paper. We would like to thank you again for your valuable time and insight to strengthen our paper.
Yours truly,
Corresponding author on behalf of the authors.
Comment |
Response to reviewer comment |
The authors have substantially improved all the changes proposed by the reviewers. This would allow the article to be published |
I greatly appreciate your kind and constructive feedback on the revised version of our manuscript. I'm glad to know that the modifications and improvements we have made in response to the comments have been received positively. It is rewarding to see our collaborative effort contributing to the enhancement of the work's quality and clarity. Your insightful suggestions played an invaluable role in refining our manuscript. The comments and recommendations provided not only helped us in rectifying the shortcomings but also enriched our understanding of the subject matter. It is our sincere hope that the paper will make a meaningful contribution to the field, thanks to your thorough review. Thank you once again for your time and effort. |

Reviewer 2 Report (Previous Reviewer 2)
The present manuscript has been much improved so that this work can be accepted now, but some detailed issues are also needed to be rectified and revised. At first, in keywords, pregnancy can be changed to pregnant women. Secondly, line 39 introduction should be deleted, and line 183 to 185 about ethics committee approval should be added to the section of research design. Thirdly, all symbol including N, t, p, F in this article should be presented with italics, but not including M, SD, Mean, variables, numbers, and significant difference from table 2 to table 7. In addition, table 6 has been cross-paged. Fourthly, the conclusion are so redundant and numerous, this section should be concise and clear, and summarize the main findings about this study, and some content can be put in the section of discussion. Last but not least, I think the authors should also compare the difference of three main variables during the same pregnant numbers between the women doing sustainable exercise and doing no exercise by the Independent Samples T-test. That's all, and thank you for authors' revision of this article. Thanks!
I strongly recommend the quality of English language should be much improved by authors. Thank you !
Author Response
Dear Reviewer,
We woldu like to thank you for the insightful comments and suggestions. We made all possible changes that were suggested and detailed the changes in the table below. Prior to response your comments we want to inform you that all the revisions and improvements are highlighted yellow in the revised version of our manuscript. We sincerely appreciate your insightful comments on our paper. We would like to thank you again for your valuable time and insight to strengthen our paper.
Yours truly,
Corresponding author on behalf of the authors.
Comment |
Response to reviewer comment |
At first, in keywords, pregnancy can be changed to pregnant women. |
Arrangements have been made according to your suggestion. |
Secondly, line 39 introduction should be deleted, and line 183 to 185 about ethics committee approval should be added to the section of research design. |
Arrangements have been made according to your suggestion. |
Thirdly, all symbol including N, t, p, F in this article should be presented with italics, but not including M, SD, Mean, variables, numbers, and significant difference from table 2 to table 7. In addition, table 6 has been cross-paged. |
Arrangements have been made according to your suggestion. |
Fourthly, the conclusion are so redundant and numerous, this section should be concise and clear, and summarize the main findings about this study, and some content can be put in the section of discussion. |
Arrangements have been made according to your suggestion. |
Last but not least, I think the authors should also compare the difference of three main variables during the same pregnant numbers between the women doing sustainable exercise and doing no exercise by the Independent Samples T-test. |
Thank you for your detailed feedback and suggestion to add a comparison of the three main variables across the same number of pregnancies between women engaging in sustainable exercise and those not exercising. Your suggestion of conducting an Independent Samples T-test is indeed a valid method to compare these groups and would generally add a valuable perspective to the findings. However, after careful consideration, I would like to clarify that the original focus of the study was to investigate the variations in self-efficacy, self-esteem, and assertiveness levels among pregnant women who engage in sustainable exercise versus those who don't, and how these differences connect to the number of pregnancies. Given the scope and design of the study, introducing a comparison as you suggested would, in fact, result in a new finding. While it could provide additional insights, it falls outside the current scope and objectives of our research, which were set and defined in the initial planning stages of our study. In future research, we will certainly take into consideration the potential benefits of examining these variables as you have suggested. For the present study, though, we believe that the findings as reported offer significant contributions to the existing body of knowledge without the need for additional analyses. We appreciate your understanding and look forward to further feedback you may have. |
I strongly recommend the quality of English language should be much improved by authors. |
Thank you for your suggestion to improve the quality of the English language in our paper. We have carefully reviewed the manuscript and made extensive revisions to the English language to improve clarity and readability. Our aim is to present our research findings as clearly as possible and we believe that the revisions reflect this intention. Changes have been made to the introduction and methods section and the revised paragraphs are shown in yellow. As the findings, discussion and conclusion sections have been previously edited, there was no need to reorganise them. We hope that these changes will meet your expectations and welcome any suggestions to improve the quality of the manuscript. We look forward to hearing from you. |

Reviewer 3 Report (Previous Reviewer 1)
I appreciate the opportunity to once again review the manuscript entitled “The Impact of Sustainable Exercise and the Number of Pregnancies on Self-Efficacy, Self-Esteem, and Assertiveness Levels in Pregnant Women” submitted in journal Sustainability.
Reviewer minor comments:
1. Please check again the grammar errors though out the manuscript
2. Please mention the aims of manuscript as well as conclusions in non structured way, no with the numbering of the paragraphs.
The authors changed the first version of the manuscript according the reviewers comments and this version is improved and my opinion is that this submission meets the criteria to be published in journal Sustainability after minor revisions I suggested.
Author Response
Dear Reviewer,
We woldu like to thank you for the insightful comments and suggestions. We made all possible changes that were suggested and detailed the changes in the table below. Prior to response your comments we want to inform you that all the revisions and improvements are highlighted yellow in the revised version of our manuscript. We sincerely appreciate your insightful comments on our paper. We would like to thank you again for your valuable time and insight to strengthen our paper.
Yours truly,
Corresponding author on behalf of the authors.
Comment |
Response to reviewer comment |
Please check again the grammar errors though out the manuscript. |
In line with your suggestion, grammar and English language quality improvements have been made extensively. The revisions cover the "Introduction and Method" sections. The paragraphs containing the edits are shown in yellow colour. Since the "Results, Discussion and Conclusion" sections have been edited before, there is no need to edit them again. |
Please mention the aims of manuscript as well as conclusions in non structured way, no with the numbering of the paragraphs. |
We appreciate your effort to provide constructive feedback to improve our manuscript. In response to your comment about the presentation of the manuscript's aims and conclusions, we have chosen to maintain the structured format for these sections to ensure clarity and readability. We believe that this structure allows for easier understanding of our main points, particularly given the complex nature of our subject matter. However, we understand the importance of making our manuscript engaging and readable, and we will take your comment into account for future manuscripts and research. We hope that the current structured presentation does not detract from the important findings of our study. We look forward to any additional feedback you may have and thank you once again for your time and consideration. |
Notes
According to other reviewer comments;
- The term pregnancy in the keywords has been changed to pregnant woman.
- The sentences related to "Ethics Committee Permission" between lines 183-185 have been placed under the title of "Research Design".
- In the previous version, M, SD, Mean, variables and numbers were presented in italics. In this version, only N, t, p and F expressions are presented in italics. Other expressions in the tables are excluded from the italicised presentation.
- In the previous version, table 6 was positioned diagonally. In this version, the position of table 6 has been changed.
- It is aimed to provide a simpler and clearer structure to the results section of the research. In this context, arrangements have been made in the results section.

This manuscript is a resubmission of an earlier submission. The following is a list of the peer review reports and author responses from that submission.
Round 1
Reviewer 1 Report
I appreciate the opportunity to review the manuscript entitled “The Impact of Sustainable Exercise and the Number of Pregnancies on Self-Efficacy, Self-Esteem, and Assertiveness Levels 3 in Pregnant Women.” by Uluöz et al. submitted in the journal Sustainability.
The authors settled a cross-sectional study aimed at evaluating the influence of sustainable exercise as well as the number of pregnancies on self-efficacy, self-esteem, and assertiveness Levels in pregnant women. The authors observed a significant difference between the mean self-efficacy and self-esteem scores of women engaged in sustainable exercise compared to those who were not. There was no significant difference between the two groups regarding assertiveness levels. The authors observed significant differences in the mean scores of self-efficacy, self-esteem, and assertiveness differed significantly in respect to the number of pregnancies, in physically active women. The authors observed significant differences in mean self-efficacy, self-esteem, and assertiveness scores in terms of the number of pregnancies in women who did not exercise.
Reviewer Comments:
1. Please prepare a list of abbreviations used in the manuscript.
2. There is a lack in information about mechanisms via physical exercise and the number of pregnancies that influence self-efficacy, self-esteem, and assertiveness scores
3. The whole manuscript lacks scientific soundness. It presents a comparison without a deeper explanation of the described results and problems. Where is demographic and clinical information of included pregnant women and their correlations with the demographic and clinical variables?
4. What is the scientific contribution of this study? The majority of references in cited in the paper are older than 5 years.
Taking into account the specific problems of the above-mentioned paper, my opinion is that
this submission does not meet the criteria to be published in the journal Sustainability.
Reviewer 2 Report
Frankly speaking, the present study is actually an interesting and attractive work, and some valuable results could be also obtained from this research. But it is a pity that some severe flaws and serious errors would be also found from the current work, as well as some detailed and normative issues. Given that, I advise this article should be refused. Thank you!
At first, due to the pregnant women as a special group, it is necessary to tell us the specific organization name about the local ethics committee, and the informed consent would be also obtained before the formal survey. Secondly, the specific test type about t-test should be presented in data analysis, namely independent samples t-test or paired samples t-test. Thirdly, all statistical symbols, such as t, F, and p should be shown with italics. Fourthly, about the data collection tools, the authors only listed three main scales, but they should also used a standard form to investigate the number of pregnancy among them, and their physical exercise condition in daily life, because the following analysis and statistics will need this data. But this tool cannot be seen by us in tools. Fifthly, the descriptive characteristics are presented in this study as frequency, mean (M), and standard deviation (SD), but this is an obvious errors in data analysis listed by the authors. In addition, all expressions about mean and standard deviation should be unified; that is to say, from table 2 to 7, X should be changed to M, and means and standard deviation should be also deleted. At last, the conclusion part should be presented separately, and not be combined with discussion section. And then, the references are not listed by the corresponding standards and requirements of the present journal, so this section should be revised and rectified carefully and seriously. By the way, the table 5 has been cross-paged, and this condition should be avoided.
Except for this detailed and normative issues, there are two serious and severe flaws existed in this study. On the one hand, the author regarded exercise with 30 minutes and twice a week as the criterion to distinguish the pregnant women who will be the sustainable exercise group and or not. So what’s the scientific basis about this? That is, why does the author want to conduct it like this? It seems no any specification or explanation in this study. On the other hand, the author intended to investigate how the levels of self-efficacy, self-esteem, and assertiveness changed in pregnant women doing exercise as compared to those doing no physical activity, so they compare the difference of three main variables on four different numbers of pregnancy for women doing sustainable exercise or doing no exercise, respectively. But actually, the author should also compare the difference of three main variables during the same pregnant numbers between the women doing sustainable exercise and doing no exercise by the Independent Samples T-test. So I think this is a serious flaw about this article.
It is highly poor of English language about this article, and some sentences and expressions are so awful.
Reviewer 3 Report
The topic of this article is quite important to promote physical activity, as it brings powerful health benefits, especially for pregnant women.
- Introduction.
The introduction is somewhat lengthy.
Reading this section should help the reader to know what is the situation of the problem posed at the time the study is conducted and to identify what is known and what is unknown about the topic under study. Period. I would not add anything more.
Halfway through reading this section it seems that I was reading a discussion, i.e., the interpretation of the results.
I propose to the authors to make an analysis and synthesis of it. Reporting the current state of the problem under study.
Another aspect, which I may not have understood well from the Spanish translation, is why call it "sustainable exercise" and "not physical activity". The difference was not clear to me.
I have read the future lines of research but...and the limitations?????. No limitations?
- Conclusion.
The presentation of the conclusions was concise. Recommendations for solving problems or improving practice should be stated, as well as specifying actions to be taken.
Reviewer 4 Report
The article should be thoroughly revised by the authors.
It has an excessively long abstract as well as an introduction that is too long for the few citations they provide. This should be improved.
What is the ethics committee that the authors have passed? They should be clearer